# Scalable Variational Inference for Dynamical Systems

**Nico S. Gorbach**[*]
Dept. of Computer Science
ETH Zurich
ngorbach@inf.ethz.ch

**Stefan Bauer**[*]
Dept. of Computer Science
ETH Zurich
bauers@inf.ethz.ch

**Joachim M. Buhmann**
Dept. of Computer Science
ETH Zurich
jbuhmann@inf.ethz.ch

## Abstract

Gradient matching is a promising tool for learning parameters and state dynamics of ordinary differential equations. It is a grid free inference approach, which, for fully observable systems is at times competitive with numerical integration. However, for many real-world applications, only sparse observations are available or even unobserved variables are included in the model description. In these cases most gradient matching methods are difficult to apply or simply do not provide satisfactory results. That is why, despite the high computational cost, numerical integration is still the gold standard in many applications. Using an existing gradient matching approach, we propose a scalable variational inference framework which can infer states and parameters simultaneously, offers computational speedups, improved accuracy and works well even under model misspecifications in a partially observable system.

## 1 Introduction

Parameter estimation for ordinary differential equations (ODE's) is challenging due to the high computational cost of numerical integration. In recent years, gradient matching techniques established themselves as successful tools [e.g. Babtie et al., 2014] to circumvent the high computational cost of numerical integration for parameter and state estimation in ordinary differential equations. Gradient matching is based on minimizing the difference between the interpolated slopes and the time derivatives of the state variables in the ODE's. First steps go back to spline based methods [Varah, 1982, Ramsay et al., 2007] where in an iterated two-step procedure coefficients and parameters are estimated. Often cubic B-splines are used as basis functions while more advanced approaches [Niu et al., 2016] use kernel functions derived from the ODE's. An overview of recent approaches with a focus on the application for systems biology is provided in Macdonald and Husmeier [2015]. It is unfortunately not straightforward to extend spline based approaches to include unobserved variables since they usually require full observability of the system. Moreover, these methods critically depend on the estimation of smoothing parameters, which are difficult to estimate when only sparse observations are available. As a solution for both problems, Gaussian process (GP) regression was proposed in Calderhead et al. [2008] and further improved in Dondelinger et al. [2013]. While both Bayesian approaches work very well for fully observable systems, they (opposite to splines) cannot simultaneously infer parameters and unobserved states and perform poorly when only combinations of variables are observed or the differential equations contain unobserved variables. Unfortunately this is the case for most practical applications [e.g. Barenco et al., 2006].

**Related work.** Archambeau et al. [2008] proposed variational inference to approximate the true process of the dynamical system by a time-varying linear system. Their approach was later signficantly extended [Ruttor et al., 2013, Ruttor and Opper, 2010, Vrettas et al., 2015]. However, similiar to [Lyons et al., 2012] they study parameter estimation in stochastic dynamical systems while our work

---

[*]The first two authors contributed equally to this work.

focuses on deterministic systems. In addition, they use the Euler-Maruyama discretization, whereas our approach is grid free. Wang and Barber [2014] propose an approach based on a belief network but as discussed in the controversy of mechanistic modelling [Macdonald et al., 2015], this leads to an intrinsic identifiability problem.

**Our contributions.** Our proposal is a scalable variational inference based framework which can infer states and parameters simultaneously, offers significant runtime improvements, improved accuracy and works well even in the case of partially observable systems. Since it is based on simplistic mean-field approximations it offers the opportunity for significant future improvements. We illustrate the potential of our work by analyzing a system of up to 1000 states in less than 400 seconds on a standard Laptop[2].

## 2  Deterministic Dynamical Systems

A deterministic dynamical system is represented by a set of $K$ ordinary differential equations (ODE's) with model parameters $\boldsymbol{\theta}$ that describe the evolution of $K$ states $\mathbf{x}(t) = [x_1(t), x_2(t), \ldots, x_K(t)]^T$ such that:

$$\dot{\mathbf{x}}(t) = \frac{d\mathbf{x}(t)}{dt} = \mathbf{f}(\mathbf{x}(t), \boldsymbol{\theta}). \tag{1}$$

A sequence of observations, $\mathbf{y}(t)$, is usually contaminated by some measurement error which we assume to be normally distributed with zero mean and variance for each of the $K$ states, i.e. $\mathbf{E} \sim \mathcal{N}(\mathbf{0}, \mathbf{D})$, with $\mathbf{D}_{ik} = \sigma_k^2 \delta_{ik}$. Thus for $N$ distinct time points the overall system may be summarized as:

$$\mathbf{Y} = \mathbf{X} + \mathbf{E}, \tag{2}$$

where

$$\mathbf{X} = [\mathbf{x}(t_1), \ldots, \mathbf{x}(t_N)] = [\mathbf{x}_1, \ldots, \mathbf{x}_K]^T, \qquad \mathbf{Y} = [\mathbf{y}(t_1), \ldots, \mathbf{y}(t_N)] = [\mathbf{y}_1, \ldots, \mathbf{y}_K]^T,$$

and $\mathbf{x}_k = [x_k(t_1), \ldots, x_k(t_N)]^T$ is the k'th state sequence and $\mathbf{y}_k = [y_k(t_1), \ldots, y_k(t_N)]^T$ are the observations. Given the observations $\mathbf{Y}$ and the description of the dynamical system (1), the aim is to estimate both state variables $\mathbf{X}$ and parameters $\boldsymbol{\theta}$. While numerical integration can be used for both problems, its computational cost is prohibitive for large systems and motivates the grid free method outlined in section 3.

## 3  GP based Gradient Matching

Gaussian process based gradient matching was originally motivated in Calderhead et al. [2008] and further developed in Dondelinger et al. [2013]. Assuming a Gaussian process prior on state variables such that:

$$p(\mathbf{X} \mid \boldsymbol{\phi}) := \prod_k \mathcal{N}(\mathbf{0}, \mathbf{C}_{\boldsymbol{\phi}_k}) \tag{3}$$

where $\mathbf{C}_{\boldsymbol{\phi}_k}$ is a covariance matrix defined by a given kernel with hyper-parameters $\boldsymbol{\phi}_k$, the $k$-th element of $\boldsymbol{\phi}$, we obtain a posterior distribution over state-variables (from (2)):

$$p(\mathbf{X} \mid \mathbf{Y}, \boldsymbol{\phi}, \boldsymbol{\sigma}) = \prod_k \mathcal{N}(\boldsymbol{\mu}_k(\mathbf{y}_k), \boldsymbol{\Sigma}_k), \tag{4}$$

where $\boldsymbol{\mu}_k(\mathbf{y}_k) := \sigma_k^{-2} \left( \sigma_k^{-2}\mathbf{I} + \mathbf{C}_{\boldsymbol{\phi}_k}^{-1} \right)^{-1} \mathbf{y}_k$ and $\boldsymbol{\Sigma}_k^{-1} := \sigma_k^{-2}\mathbf{I} + \mathbf{C}_{\boldsymbol{\phi}_k}^{-1}$.

Assuming that the covariance function $\mathbf{C}_{\boldsymbol{\phi}_k}$ is differentiable and using the closure property under differentiation of Gaussian processes, the conditional distribution over state derivatives is:

$$p(\dot{\mathbf{X}} \mid \mathbf{X}, \boldsymbol{\phi}) = \prod_k \mathcal{N}(\dot{\mathbf{x}}_k \mid \mathbf{m}_k, \mathbf{A}_k), \tag{5}$$

where the mean and covariance is given by:

$$\mathbf{m}_k := {}'\mathbf{C}_{\phi_k}\mathbf{C}_{\phi_k}^{-1}\mathbf{x}_k, \quad \mathbf{A}_k := \mathbf{C}''_{\phi_k} - {}'\mathbf{C}_{\phi_k}\mathbf{C}_{\phi_k}^{-1}\mathbf{C}'_{\phi_k}, \tag{6}$$

$\mathbf{C}''_{\phi_k}$ denotes the auto-covariance for each state-derivative with $\mathbf{C}'_{\phi_k}$ and ${}'\mathbf{C}_{\phi_k}$ denoting the cross-covariances between the state and its derivative.

Assuming additive, normally distributed noise with state-specific error variance $\gamma_k$ in (1), we have:

$$p(\dot{\mathbf{X}} \mid \mathbf{X}, \boldsymbol{\theta}, \boldsymbol{\gamma}) = \prod_k \mathcal{N}\left(\dot{\mathbf{x}}_k \mid \mathbf{f}_k(\mathbf{X}, \boldsymbol{\theta}), \gamma_k \mathbf{I}\right). \tag{7}$$

A product of experts approach, combines the ODE informed distribution of state-derivatives (distribution (7)) with the smoothed distribution of state-derivatives (distribution (5)):

$$p(\dot{\mathbf{X}} \mid \mathbf{X}, \boldsymbol{\theta}, \boldsymbol{\phi}, \boldsymbol{\gamma}) \propto p(\dot{\mathbf{X}} \mid \mathbf{X}, \boldsymbol{\phi}) p(\dot{\mathbf{X}} \mid \mathbf{X}, \boldsymbol{\theta}, \boldsymbol{\gamma}) \tag{8}$$

The motivation for the product of experts is that the multiplication implies that both the data fit and the ODE response have to be satisfied at the same time in order to achieve a high value of $p(\dot{\mathbf{X}} \mid \mathbf{X}, \boldsymbol{\theta}, \boldsymbol{\phi}, \boldsymbol{\gamma})$. This is contrary to a mixture model, i.e. a normalized addition, where a high value for one expert e.g. overfitting the data while neglecting the ODE response or vice versa, is acceptable.

The proposed methodology in Calderhead et al. [2008] is to analytically integrate out $\dot{\mathbf{X}}$:

$$\begin{aligned}
p(\boldsymbol{\theta}|\mathbf{X}, \boldsymbol{\phi}, \boldsymbol{\gamma}) &= Z_{\boldsymbol{\theta}}^{-1}(\mathbf{X})\, p(\boldsymbol{\theta}) \int p(\dot{\mathbf{X}}|\mathbf{X}, \boldsymbol{\phi}) p(\dot{\mathbf{X}}|\mathbf{X}, \boldsymbol{\theta}, \boldsymbol{\gamma}) d\dot{\mathbf{X}} \\
&= Z_{\boldsymbol{\theta}}^{-1}(\mathbf{X})\, p(\boldsymbol{\theta}) \prod_k \mathcal{N}(\mathbf{f}_k(\mathbf{X}, \boldsymbol{\theta})|\mathbf{m}_k, \boldsymbol{\Lambda}_k^{-1}),
\end{aligned} \tag{9}$$

with $\boldsymbol{\Lambda}_k^{-1} := \mathbf{A}_k + \gamma_k \mathbf{I}$ and $Z_{\boldsymbol{\theta}}^{-1}(\mathbf{X})$ as the normalization that depends on the states $\mathbf{X}$. Calderhead et al. [2008] infer the parameters $\boldsymbol{\theta}$ by first sampling the states (i.e. $\mathbf{X} \sim p(\mathbf{X} \mid \mathbf{Y}, \boldsymbol{\phi}, \boldsymbol{\sigma})$) followed by sampling the parameters given the states (i.e. $\boldsymbol{\theta}, \boldsymbol{\gamma} \sim p(\boldsymbol{\theta}, \boldsymbol{\gamma} \mid \mathbf{X}, \boldsymbol{\phi}, \boldsymbol{\sigma})$). In this setup, sampling $\mathbf{X}$ is independent of $\boldsymbol{\theta}$, which implies that $\boldsymbol{\theta}$ and $\boldsymbol{\gamma}$ have no influence on the inference of the state variables. The desired feedback loop was closed by Dondelinger et al. [2013] through sampling from the joint posterior of $p(\boldsymbol{\theta} \mid \mathbf{X}, \boldsymbol{\phi}, \boldsymbol{\sigma}, \boldsymbol{\gamma}, \mathbf{Y})$. Since sampling the states only provides their values at discrete time points, Calderhead et al. [2008] and Dondelinger et al. [2013] require the existence of an external ODE solver to obtain continuous trajectories of the state variables. For simplicity, we derived the approach assuming full observability. However, the approach has the advantage (as opposed to splines) that the assumption of full observability can be relaxed to include only observations for combinations of states by replacing (2) with $\mathbf{Y} = \mathbf{A}\mathbf{X} + \mathbf{E}$, where $\mathbf{A}$ encodes the linear relationship between observations and states. In addition, unobserved states can be naturally included in the inference by simply using the prior on state variables (3) [Calderhead et al., 2008].

## 4 Variational Inference for Gradient Matching by Exploiting Local Linearity in ODE's

For subsequent sections we consider only models of the form (1) with reactions based on mass-action kinetics which are given by:

$$f_k(\mathbf{x}(t), \boldsymbol{\theta}) = \sum_{i=1} \theta_{ki} \prod_{j \in \mathcal{M}_{ki}} x_j \tag{10}$$

with $\mathcal{M}_{ki} \subseteq \{1, \ldots, K\}$ describing the state variables in each factor of the equation i.e. the functions are linear in parameters and contain arbitrary large products of monomials of the states. The motivation for the restriction to this functional class is twofold. First, this formulation includes models which exhibit periodicity as well as high nonlinearity and especially physically realistic reactions in systems biology [Schillings et al., 2015].

Second, the true joint posterior over all unknowns is given by:

$$p(\boldsymbol{\theta}, \mathbf{X} \mid \mathbf{Y}, \boldsymbol{\phi}, \boldsymbol{\gamma}, \boldsymbol{\sigma}) = p(\boldsymbol{\theta} \mid \mathbf{X}, \boldsymbol{\phi}, \boldsymbol{\gamma}) p(\mathbf{X} \mid \mathbf{Y}, \boldsymbol{\phi}, \boldsymbol{\sigma})$$
$$= Z_{\boldsymbol{\theta}}^{-1}(\mathbf{X}) \, p(\boldsymbol{\theta}) \prod_k \mathcal{N}\left(\mathbf{f}_k(\mathbf{X}, \boldsymbol{\theta}) \mid \mathbf{m}_k, \boldsymbol{\Lambda}_k^{-1}\right) \mathcal{N}\left(\mathbf{x}_k \mid \boldsymbol{\mu}_k(\mathbf{Y}), \boldsymbol{\Sigma}_k\right),$$

where the normalization of the parameter posterior (9), $Z_{\boldsymbol{\theta}}(\mathbf{X})$, depends on the states $\mathbf{X}$. The dependence is nontrivial and induced by the nonlinear couplings of the states $\mathbf{X}$, which make the inference (e.g. by integration) challenging in the first place. Previous approaches ignore the dependence of $Z_{\boldsymbol{\theta}}(\mathbf{X})$ on the states $\mathbf{X}$ by setting $Z_{\boldsymbol{\theta}}(\mathbf{X})$ equal to one [Dondelinger et al., 2013, equation 20]. We determine $Z_{\boldsymbol{\theta}}(\mathbf{X})$ analytically by exploiting the *local* linearity of the ODE's as shown in section 4.1 (and section 7 in the supplementary material). More precisely, for mass action kinetics 10, we can rewrite the ODE's as a linear combination in an *individual* state or as a linear combination in the ODE parameters[3]. We thus achieve superior performance over existing gradient matching approaches, as shown in the experimental section 5.

## 4.1 Mean-field Variational Inference

To infer the parameters $\boldsymbol{\theta}$, we want to find the maximum *a posteriori* estimate (MAP):

$$\boldsymbol{\theta}^\star := \underset{\theta}{\operatorname{argmax}} \ln p(\boldsymbol{\theta} \mid \mathbf{Y}, \boldsymbol{\phi}, \boldsymbol{\gamma}, \boldsymbol{\sigma}) = \underset{\theta}{\operatorname{argmax}} \ln \int \underbrace{p(\boldsymbol{\theta} \mid \mathbf{X}, \boldsymbol{\phi}, \boldsymbol{\gamma}) p(\mathbf{X} \mid \mathbf{Y}, \boldsymbol{\phi}, \boldsymbol{\sigma})}_{=p(\boldsymbol{\theta}, \mathbf{X} \mid \mathbf{Y}, \boldsymbol{\phi}, \boldsymbol{\gamma}, \boldsymbol{\sigma})} d\mathbf{X} \quad (11)$$

However, the integral in (11) is intractable in most cases due to the strong couplings induced by the nonlinear ODE's $\mathbf{f}$ which appear in the term $p(\boldsymbol{\theta} \mid \mathbf{X}, \boldsymbol{\phi}, \boldsymbol{\gamma})$ (equation 9). We therefore use mean-field variational inference to establish variational lower bounds that are analytically tractable by decoupling state variables from the ODE parameters as well as decoupling the state variables from each other. Before explaining the mechanism behind mean-field variational inference, we first observe that, due to the model assumption (10), the true conditional distributions $p(\boldsymbol{\theta} \mid \mathbf{X}, \mathbf{Y}, \boldsymbol{\phi}, \boldsymbol{\gamma}, \boldsymbol{\sigma})$ and $p(\mathbf{x}_u \mid \boldsymbol{\theta}, \mathbf{X}_{-u}, \mathbf{Y}, \boldsymbol{\phi}, \boldsymbol{\gamma}, \boldsymbol{\sigma})$ are Gaussian distributed, where $\mathbf{X}_{-u}$ denotes all states excluding state $\mathbf{x}_u$ (i.e. $\mathbf{X}_{-u} := \{\mathbf{x} \in \mathbf{X} \mid \mathbf{x} \neq \mathbf{x}_u\}$). For didactical reasons, we write the true conditional distributions in canonical form:

$$p(\boldsymbol{\theta} \mid \mathbf{X}, \mathbf{Y}, \boldsymbol{\phi}, \boldsymbol{\gamma}, \boldsymbol{\sigma}) = h(\boldsymbol{\theta}) \times \exp\left(\boldsymbol{\eta}_{\boldsymbol{\theta}}(\mathbf{X}, \mathbf{Y}, \boldsymbol{\phi}, \boldsymbol{\gamma}, \boldsymbol{\sigma})^T \mathbf{t}(\boldsymbol{\theta}) - a_{\boldsymbol{\theta}}(\boldsymbol{\eta}_{\boldsymbol{\theta}}(\mathbf{X}, \mathbf{Y}, \boldsymbol{\phi}, \boldsymbol{\gamma}, \boldsymbol{\sigma}))\right)$$
$$p(\mathbf{x}_u \mid \boldsymbol{\theta}, \mathbf{X}_{-u}, \mathbf{Y}, \boldsymbol{\phi}, \boldsymbol{\gamma}, \boldsymbol{\sigma}) = h(\mathbf{x}_u) \times \exp\left(\boldsymbol{\eta}_u(\boldsymbol{\theta}, \mathbf{X}_{-u}, \mathbf{Y}, \boldsymbol{\phi}, \boldsymbol{\gamma}, \boldsymbol{\sigma})^T \mathbf{t}(\mathbf{x}_u)\right.$$
$$\left. - a_u(\boldsymbol{\eta}_u(\mathbf{X}_{-u}, \mathbf{Y}, \boldsymbol{\phi}, \boldsymbol{\gamma}, \boldsymbol{\sigma}))\right) \quad (12)$$

where $h(\cdot)$ and $a(\cdot)$ are the base measure and log-normalizer and $\boldsymbol{\eta}(\cdot)$ and $\mathbf{t}(\cdot)$ are the natural parameter and sufficient statistics.

The decoupling is induced by designing a variational distribution $Q(\boldsymbol{\theta}, \mathbf{X})$ which is restricted to the family of factorial distributions:

$$\mathcal{Q} := \left\{ Q : Q(\boldsymbol{\theta}, \mathbf{X}) = q(\boldsymbol{\theta} \mid \boldsymbol{\lambda}) \prod_u q(\mathbf{x}_u \mid \boldsymbol{\psi}_u) \right\}, \quad (13)$$

where $\boldsymbol{\lambda}$ and $\boldsymbol{\psi}_u$ are the variational parameters. The particular form of $q(\boldsymbol{\theta} \mid \boldsymbol{\lambda})$ and $q(\mathbf{x}_u \mid \boldsymbol{\psi}_u)$ is designed to be in the same exponential family as the true conditional distributions in equation (12):

$$q(\boldsymbol{\theta} \mid \boldsymbol{\lambda}) := h(\boldsymbol{\theta}) \exp\left(\boldsymbol{\lambda}^T \mathbf{t}(\boldsymbol{\theta}) - a_{\boldsymbol{\theta}}(\boldsymbol{\lambda})\right)$$
$$q(\mathbf{x}_u \mid \boldsymbol{\psi}_u) := h(\mathbf{x}_u) \exp\left(\boldsymbol{\psi}_u^T \mathbf{t}(\mathbf{x}_u) - a_u(\boldsymbol{\psi}_u)\right)$$

To find the optimal factorial distribution we minimize the Kullback-Leibler divergence between the variational and the true posterior distribution:

$$\hat{Q} := \operatorname*{argmin}_{Q(\boldsymbol{\theta}, \mathbf{X}) \in \mathcal{Q}} \mathrm{KL}\left[Q(\boldsymbol{\theta}, \mathbf{X}) \| p(\boldsymbol{\theta}, \mathbf{X} \mid \mathbf{Y}, \boldsymbol{\phi}, \boldsymbol{\gamma}, \boldsymbol{\sigma})\right]$$

$$= \operatorname*{argmin}_{Q(\boldsymbol{\theta}, \mathbf{X}) \in \mathcal{Q}} \mathbb{E}_Q \log Q(\boldsymbol{\theta}, \mathbf{X}) - \mathbb{E}_Q \log p(\boldsymbol{\theta}, \mathbf{X} \mid \mathbf{Y}, \boldsymbol{\phi}, \boldsymbol{\gamma}, \boldsymbol{\sigma})$$

$$= \operatorname*{argmax}_{Q(\boldsymbol{\theta}, \mathbf{X}) \in \mathcal{Q}} \mathcal{L}_Q(\boldsymbol{\lambda}, \boldsymbol{\psi}) \tag{14}$$

where $\hat{Q}$ is the proxy distribution and $\mathcal{L}_Q(\boldsymbol{\lambda}, \boldsymbol{\psi})$ is the ELBO (Evidence Lower Bound) terms that depends on the variational parameters $\boldsymbol{\lambda}$ and $\boldsymbol{\psi}$. Maximizing ELBO w.r.t. $\boldsymbol{\theta}$ is equivalent to maximizing the following lower bound:

$$\mathcal{L}_{\boldsymbol{\theta}}(\boldsymbol{\lambda}) := \mathbb{E}_Q \log p(\boldsymbol{\theta} \mid \mathbf{X}, \mathbf{Y}, \boldsymbol{\phi}, \boldsymbol{\gamma}, \boldsymbol{\sigma}) - \mathbb{E}_Q \log q(\boldsymbol{\theta} \mid \boldsymbol{\lambda})$$

$$= \mathbb{E}_Q \boldsymbol{\eta}_{\boldsymbol{\theta}}^T \triangledown_{\boldsymbol{\lambda}} a_{\boldsymbol{\theta}}(\boldsymbol{\lambda}) - \boldsymbol{\lambda}^T \triangledown_{\boldsymbol{\lambda}} a_{\boldsymbol{\theta}}(\boldsymbol{\lambda}),$$

where we substitute the true conditionals given in equation (12) and $\triangledown_{\boldsymbol{\lambda}}$ is the gradient operator. Similarly, maximizing ELBO w.r.t. latent state $\mathbf{x}_u$, we have:

$$\mathcal{L}_x(\boldsymbol{\psi}_u) := \mathbb{E}_Q \log p(\mathbf{x}_u \mid \boldsymbol{\theta}, \mathbf{X}_{-u}, \mathbf{Y}, \boldsymbol{\phi}, \boldsymbol{\gamma}, \boldsymbol{\sigma}) - \mathbb{E}_Q \log q(\mathbf{x}_u \mid \boldsymbol{\psi}_u)$$

$$= \mathbb{E}_Q \boldsymbol{\eta}_u^T \triangledown_{\boldsymbol{\psi}_u} a_u(\boldsymbol{\psi}_u) - \boldsymbol{\psi}_u^T \triangledown_{\boldsymbol{\psi}_u} a_u(\boldsymbol{\psi}_u)$$

Given the assumptions we made about the true posterior and the variational distribution (i.e. that each true conditional is in an exponential family and that the corresponding variational distribution is in the same exponential family) we can optimize each coordinate in closed form.

To maximize ELBO we set the gradient w.r.t. the variational parameters to zero:

$$\triangledown_{\boldsymbol{\lambda}} \mathcal{L}_{\boldsymbol{\theta}}(\boldsymbol{\lambda}) = \triangledown_{\boldsymbol{\lambda}}^2 a_{\boldsymbol{\theta}}(\boldsymbol{\lambda}) \left(\mathbb{E}_Q \boldsymbol{\eta}_{\boldsymbol{\theta}} - \boldsymbol{\lambda}\right) \stackrel{!}{=} 0$$

which is zero when:

$$\hat{\boldsymbol{\lambda}} = \mathbb{E}_Q \boldsymbol{\eta}_{\boldsymbol{\theta}} \tag{15}$$

Similarly, the optimal variational parameters of the states are given by:

$$\hat{\boldsymbol{\psi}}_u = \mathbb{E}_Q \boldsymbol{\eta}_u \tag{16}$$

Since the true conditionals are Gaussian distributed the expectations over the natural parameters are given by:

$$\mathbb{E}_Q \boldsymbol{\eta}_{\boldsymbol{\theta}} = \begin{pmatrix} \mathbb{E}_Q \boldsymbol{\Omega}_{\boldsymbol{\theta}}^{-1} \mathbf{r}_{\boldsymbol{\theta}} \\ -\frac{1}{2} \mathbb{E}_Q \boldsymbol{\Omega}_{\boldsymbol{\theta}}^{-1} \end{pmatrix}, \quad \mathbb{E}_Q \boldsymbol{\eta}_u = \begin{pmatrix} \mathbb{E}_Q \boldsymbol{\Omega}_u^{-1} \mathbf{r}_u \\ -\frac{1}{2} \mathbb{E}_Q \boldsymbol{\Omega}_u^{-1} \end{pmatrix}, \tag{17}$$

where $\mathbf{r}_{\boldsymbol{\theta}}$ and $\boldsymbol{\Omega}_{\boldsymbol{\theta}}$ are the mean and covariance of the true conditional distribution over ODE parameters. Similarly, $\mathbf{r}_u$ and $\boldsymbol{\Omega}_u$ are the mean and covariance of the true conditional distribution over states. The variational parameters in equation (17) are derived *analytically* in the supplementary material 7.

The coordinate ascent approach (where each step is analytically tractable) for estimating states and parameters is summarized in algorithm 1.

---

**Algorithm 1** Mean-field coordinate ascent for GP Gradient Matching

1: Initialization of proxy moments $\boldsymbol{\eta}_u$ and $\boldsymbol{\eta}_{\boldsymbol{\theta}}$.
2: **repeat**
3:     Given the proxy over ODE parameters $q(\boldsymbol{\theta} \mid \hat{\boldsymbol{\lambda}})$, calculate the proxy over individual states $q(\mathbf{x}_u \mid \hat{\boldsymbol{\psi}}_u) \, \forall \, u \leq n$, by computing its moments $\hat{\boldsymbol{\psi}}_u = \mathbb{E}_Q \boldsymbol{\eta}_u$.
4:     Given the proxy over individual states $q(\mathbf{x}_u \mid \hat{\boldsymbol{\psi}}_u)$, calculate the proxy over ODE parameters $q(\boldsymbol{\theta} \mid \hat{\boldsymbol{\lambda}})$, by computing its moments $\hat{\boldsymbol{\lambda}} = \mathbb{E}_Q \boldsymbol{\eta}_{\boldsymbol{\theta}}$.
5: **until** convergence of maximum number of iterations is exceeded.

---

Assuming that the maximal number of states for each equation in (10) is constant (which is to the best of our knowledge the case for any reasonable dynamical system), the computational complexity of the algorithm is *linear* in the states $\mathcal{O}(N \cdot K)$ for each iteration. This result is experimentally supported by figure 5 where we analyzed a system of up to 1000 states in less than 400 seconds.

# 5 Experiments

In order to provide a fair comparison to existing approaches, we test our approach on two small to medium sized ODE models, which have been extensively studied in the same parameter settings before [e.g. Calderhead et al., 2008, Dondelinger et al., 2013, Wang and Barber, 2014]. Additionally, we show the scalability of our approach on a large-scale partially observable system which has so far been infeasible to analyze with existing gradient matching methods due to the number of unobserved states.

## 5.1 Lotka-Volterra

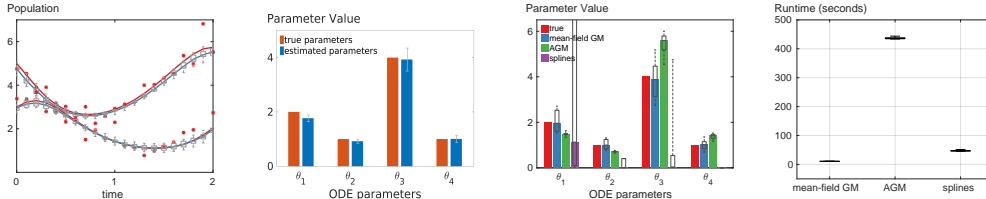

Figure 1: **Lotka-Volterra**: Given few noisy observations (red stars), simulated with a variance of $\sigma^2 = 0.25$, the leftmost plot shows the inferred state dynamics using our variational mean-field method (mean-field GM, median runtime 4.7sec). Estimated mean and standard deviation for one random data initialization using our approach are illustrated in the left-center plot. The implemented spline method (splines, median runtime 48sec) was based on Niu et al. [2016] and the adaptive gradient matching (AGM) is the approach proposed by Dondelinger et al. [2013]. Boxplots in the leftmost, right-center and rightmost plot illustrate the variance in the state and parameter estimations over 10 independent datasets.

The ODE's $\mathbf{f}(\mathbf{X}, \boldsymbol{\theta})$ of the Lotka-Volterra system [Lotka, 1978] is given by:

$$\dot{x}_1 := \theta_1 x_1 - \theta_2 x_1 x_2$$
$$\dot{x}_2 := -\theta_3 x_2 + \theta_4 x_1 x_2$$

The above system is used to study predator-prey interactions and exhibits periodicity and non-linearity at the same time. We used the same ODE parameters as in Dondelinger et al. [2013] (i.e. $\theta_1 = 2, \theta_2 = 1, \theta_3 = 4, \theta_4 = 1$) to simulate the data over an interval $[0, 2]$ with a sampling interval of $0.1$. Predator species (i.e. $x_1$) were initialized to 3 and prey species (i.e. $x$) were initialized to 5. Mean-field variational inference for gradient matching was performed on a simulated dataset with additive Gaussian noise with variance $\sigma^2 = 0.25$. The radial basis function kernel was used to capture the covariance between a state at different time points.

As shown in figure 1, our method performs significantly better than all other methods at a fraction of the computational cost. The poor performance in accuracy of Niu et al. [2016] can be explained by the significantly lower number of samples and higher noise level, compared to the simpler setting of their experiments. In order to show the potential of our work we decided to follow the more difficult and established experimental settings used in [e.g. Calderhead et al., 2008, Dondelinger et al., 2013, Wang and Barber, 2014]. This illustrates the difficulty of

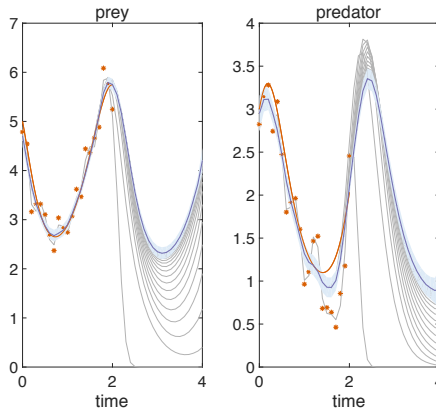

Figure 2: **Lotka-Volterra**: Given only observations (red stars) until time $t = 2$ the state trajectories are inferred including the unobserved time points up to time $t = 4$. The typical patterns of the Lotka-Volterra system for predator and prey species are recovered. The shaded blue area shows the uncertainty around for the inferred state trajectories.

spline based gradient matching methods when only few observations are available. We estimated the smoothing parameter $\lambda$ in the proposal of Niu et al. [2016] using leave-one-out cross-validation. While their method can in principle achieve the same runtime (e.g. using 10-fold cv) as our method, the performance for parameter estimation is significantly worse already when using leave-one-out cross-validation, where the median parameter estimation over ten independent data initializations is completely off for three out of four parameters (figure 1). Adaptive gradient matching (AGM) [Dondelinger et al., 2013] would eventually converge to the true parameter values but at roughly 100 times the runtime achieves signifcantly worse results in accuracy than our approach (figure 1). In figure 2 we additionally show that the mechanism of the Lotka-Volterra system is correctly inferred even when including unobserved time intervals.

## 5.2    Protein Signalling Transduction Pathway

In the following we only compare with the current state of the art in GP based gradient matching [Dondelinger et al., 2013] since spline methods are in general difficult or inapplicable for partial observable systems. In addition, already in the case of a simpler system and more data points (e.g. figure 1), splines were not competitive (in accuracy) with the approach of Dondelinger et al. [2013].

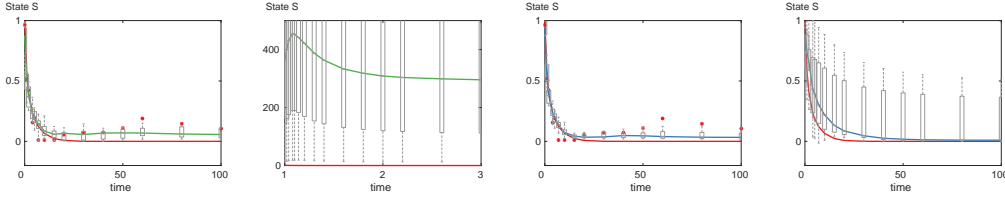

Figure 3: For the noise level of $\sigma^2 = 0.1$ the leftmost and left-center plot show the performance of Dondelinger et al. [2013](AGM) for inferring the state trajectories of state S. The red curve in all plots is the groundtruth, while the inferred trajectories of AGM are plotted in green (left and left-center plot) and in blue (right and right center) for our approach. While in the scenario of the leftmost and right-center plot observations are available (red stars) and both approaches work well, the approach of Dondelinger et al. [2013](AGM) is significantly off in inferring the same state when it is unobserved but all other parameters remain the same (left-center plot) while our approach infers similar dynamics in both scenarios.

The chemical kinetics for the protein signalling transduction pathway is governed by a combination of mass action kinetics and the Michaelis-Menten kinetics:

$$\dot{S} = -k_1 \times S - k_2 \times S \times R + k_3 \times RS$$
$$\dot{dS} = k_1 \times S$$
$$\dot{R} = -k_2 \times S \times R + k_3 \times RS + V \times \frac{Rpp}{K_m + Rpp}$$
$$\dot{RS} = k_2 \times S \times R - k_3 \times RS - k_4 \times RS$$
$$\dot{Rpp} = k_4 \times RS - V \times \frac{Rpp}{K_m + Rpp}$$

For a detailed descripton of the systems with its biological interpretations we refer to Vyshemirsky and Girolami [2008]. While mass-action kinetics in the protein transduction pathway satisfy our constraints on the functional form of the ODE's 1, the Michaelis-Menten kinetics do not, since they give rise to the ratio of states $\frac{Rpp}{K_m + Rpp}$. We therefore define the following latent variables:

$$x_1 := S, \ x_2 := dS, \ x_3 := R, \ x_4 := RS, \ x_5 := \frac{Rpp}{K_m + Rpp}$$
$$\theta_1 := k_1, \theta_2 := k_2, \theta_3 := k_3, \theta_4 := k_4, \theta_5 := V$$

The transformation is motivated by the fact that in the new system, all states only appear as monomials, as required in (10). Our variable transformation includes an inherent error (e.g. by replacing

$\dot{Rpp} = k_4 \times RS - V \times \frac{Rpp}{K_m + Rpp}$ with $\dot{x}_5 = \theta_4 \times x_4 - \theta_5 \times x_5$) but despite such a misspecification, our method estimates four out of five parameters correctly (4). Once more, we use the same ODE parameters as in Dondelinger et al. [2013] i.e. $k_1 = 0.07, k_2 = 0.6, k_3 = 0.05, k_4 = 0.3, V = 0.017$. The data was sampled over an interval $[0, 100]$ with time point samples at $t = [0, 1, 2, 4, 5, 7, 10, 15, 20, 30, 40, 50, 60, 80, 100]$. Parameters were inferred in two experiments with different standard Gaussian distributed noise with variances $\sigma^2 = 0.01$ and $\sigma^2 = 0.1$.

Even for a misspecified model, containing a systematic error, the ranking according to parameter values is preserved as indicated in figure 4. While the approach of Dondelinger et al. [2013] converges much slower (again factor 100 in runtime) to the true values of the parameters (for a fully observable system), it is significantly off if state S is unobserved and is more sensitive to the introduction of noise than our approach (figure 3). Our method infers similar dynamics for the fully and partially observable system as shown in figure 3 and remains unchanged in its estimation accuracy after the introduction of unobserved variables (even having its inherent bias) and performs well even in comparison to numerical integration (figure 4). Plots for the additional state dynamics are shown in the supplementary material 6.

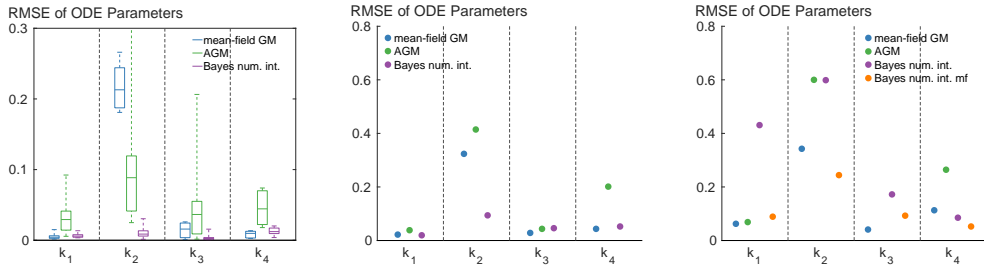

Figure 4: From the left to the right the plots represent three different inference settings of increasing difficulty using the protein transduction pathway as an example. The left plot shows the results for a fully observable system and a small noise level ($\sigma^2 = 0.01$). Due to the violation of the functional form assumption our approach has an inherent bias and Dondelinger et al. [2013](AGM) performs better while Bayesian numerical integration (Bayes num. int.) serves as a gold standard and performs best. The middle plot shows the same system with an increased noise level of $\sigma^2 = 0.1$. Due to many outliers we only show the median over ten independent runs and adjust the scale for the middle and right plot. In the right plot state S was unobserved while the noise level was kept at $\sigma^2 = 0.1$ (the estimate for $k_3$ of AGM is at 18 and out of the limits of the plot). Initializing numerical integration with our result (Bayes num. int. mf.) achieves the best results and significantly lowers the estimation error (right plot).

## 5.3 Scalability

To show the true scalability of our approach we apply it to the Lorenz 96 system, which consists of equations of the form:

$$f_k(\mathbf{x}(t), \boldsymbol{\theta}) = (x_{k+1} - x_{k-2})x_{k-1} - x_k + \theta, \tag{18}$$

where $\theta$ is a scalar forcing parameter, $x_{-1} = x_{K-1}, x_0 = x_K$ and $x_{K+1} = x_1$ (with $K$ being the number of states in the deterministic system (1)). The Lorenz 96 system can be seen as a minimalistic weather model [Lorenz and Emanuel, 1998] and is often used with an additional diffusion term as a reference model for stochastic systems [e.g. Vrettas et al., 2015]. It offers a flexible framework for increasing the number states in the inference problem and in our experiments we use between 125 to 1000 states. Due to the dimensionality the Lorenz 96 system has so far not been analyzed using gradient matching methods and to additionally increase the difficulty of the inference problem we randomly selected *one third* of the states to be unobserved. We simulated data setting $\theta = 8$ with an observation noise of $\sigma^2 = 1$ using 32 equally space observations between zero to four seconds. Due to its scaling properties, our approach is able to infer a system with 1000 states within less than 400 seconds (right plot in figure 5). We can visually conclude that unobserved states are approximately correct inferred and the approximation error is independent of the dimensionality of the problem (right plot in figure 5).

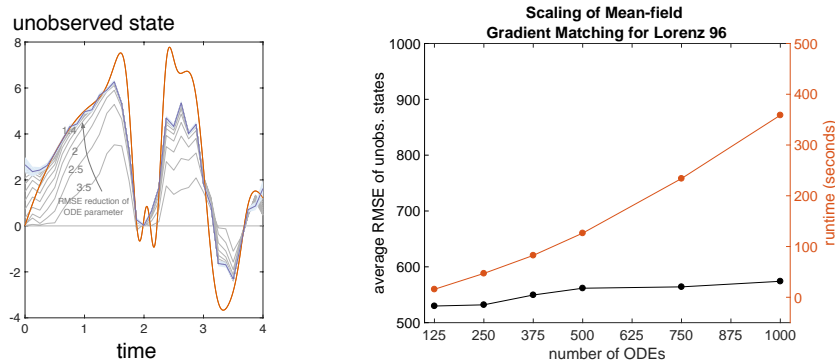

Figure 5: The left plot shows the improved mechanistic modelling and the reduction of the root median squared error (RMSE) with each iteration of our algorithm. The groundtruth for an unobserved state is plotted in red while the thin gray lines correspond to the inferred state trajectories in each iteration of the algorithm (the first flat thin gray line being the initialisation). The blue line is the inferred state trajectory of the unobserved state after convergence. The right plot shows the scaling of our algorithm with the dimensionality in the states. The red curve is the runtime in seconds wheras the blue curve is corresponding to the RSME (right plot).

Due to space limitations, we show additional experiments for various dynamical systems in the fields of fluid dynamics, electrical engineering, system biology and neuroscience only in the supplementary material in section 8.

## 6 Discussion

Numerical integration is a major bottleneck due to its computational cost for large scale estimation of parameters and states e.g. in systems biology. However, it still serves as the gold standard for practical applications. Techniques based on gradient matching offer a computationally appealing and successful shortcut for parameter inference but are difficult to extend to include unobserved variables in the model descripton or are unable to keep their performance level from fully observed systems. However, most real world applications are only partially observed. Provided that state variables appear as monomials in the ODE, we offer a simple, yet powerful inference framework that is scalable, significantly outperforms existing approaches in runtime and accuracy and performs well in the case of sparse observations even for partially observable systems. Many non-linear and periodic ODE's, e.g. the Lotka-Volterra system, already fulfill our assumptions. The empirically shown robustness of our model to misspecification even in the case of additional partial observability already indicates that a relaxation of the functional form assumption might be possible in future research.

## Acknowledgements

This research was partially supported by the Max Planck ETH Center for Learning Systems and the SystemsX.ch project SignalX.

## Footnotes

[2]All experiments were run on a 2.5 GHz Intel Core i7 Macbook.

[3]For mass-action kinetics as in (10), the ODE's are nonlinear in all states but linear in a single state as well as linear in all ODE parameters.

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
