[Supplementary Material]

# Supplementary Material for:
# "Scalable Variational Inference for Dynamical Systems"

We start by deriving the variational parameters for mean-field gradient matching in section 7 followed by a comparison of methods to estimate parameters and states of the protein signalling transduction pathway in section 8.1. The subsequent sections demonstrate mean-field gradient matching for various dynamical systems in the fields of fluid dynamics (i.e. Lorenz 96 system, section 8.2), electrical engineering (i.e. FitzHugh-Nagumo system, section 8.3), system biology (i.e. glucose uptake into yeast, section 8.4) and neuroscience (i.e. dynamic causal modelling, section 8.4.1). Lastly, we compare the different kernels used for the estimation of each system in section 8.4.2.

## 7  Variational Parameters

The true conditional over ODE parameters is given by:

$$
\begin{aligned}
p(\boldsymbol{\theta} \mid \mathbf{X}, \mathbf{Y}, \boldsymbol{\phi}, \boldsymbol{\gamma}, \boldsymbol{\sigma}) &\overset{(a)}{=} p(\boldsymbol{\theta} \mid \mathbf{X}, \boldsymbol{\phi}, \boldsymbol{\gamma}) \\
&\overset{(b)}{=} Z_{\boldsymbol{\theta}}^{-1}(\mathbf{X}) \int p(\dot{\mathbf{X}} \mid \mathbf{X}, \boldsymbol{\theta}, \boldsymbol{\phi}, \boldsymbol{\gamma}) p(\dot{\mathbf{X}} \mid \mathbf{X}, \boldsymbol{\phi}) d\dot{\mathbf{X}} \\
&\overset{(c)}{=} Z_{\boldsymbol{\theta}}^{-1}(\mathbf{X}) \prod_k \mathcal{N}\left(\mathbf{f}_k(\mathbf{X}, \boldsymbol{\theta}) \mid \mathbf{m}_k, \boldsymbol{\Lambda}_k^{-1}\right) \\
&\overset{(d)}{=} Z_{\boldsymbol{\theta}}^{-1}(\mathbf{X}) \prod_k \mathcal{N}\left(\mathbf{B}_{\boldsymbol{\theta} k} \boldsymbol{\theta} + \mathbf{b}_{\boldsymbol{\theta} k} \mid \mathbf{m}_k, \boldsymbol{\Lambda}_k^{-1}\right) \\
&\overset{(e)}{=} Z_{\boldsymbol{\theta}}^{\prime -1}(\mathbf{X}) \prod_k \mathcal{N}\left(\boldsymbol{\theta} \mid \left(\mathbf{B}_{\boldsymbol{\theta} k}^T \boldsymbol{\Lambda}_k \mathbf{B}_{\boldsymbol{\theta} k}\right)^{-1} \mathbf{B}_{\boldsymbol{\theta} k}^T \boldsymbol{\Lambda}_k (\mathbf{m}_k - \mathbf{f}_{\boldsymbol{\theta} k}), \left(\mathbf{B}_{\boldsymbol{\theta} k}^T \boldsymbol{\Lambda}_k \mathbf{B}_{\boldsymbol{\theta} k}\right)^{-1}\right) \\
&\overset{(f)}{=} \mathcal{N}\left(\boldsymbol{\theta} \mid \mathbf{r}_{\boldsymbol{\theta}}, \boldsymbol{\Omega}_{\boldsymbol{\theta}}\right),
\end{aligned}
$$

where $Z_{\boldsymbol{\theta}}^{-1}(\mathbf{X})$ and $Z_{\boldsymbol{\theta}}^{\prime -1}(\mathbf{X})$ normalize the distributions and $\mathbf{m}_k$ and $\boldsymbol{\Lambda}_k$ are defined in equations (6) and (9), respectively. In (a) we notice that $\boldsymbol{\theta}$ does not directly depend on the observations $\mathbf{Y}$ but instead indirectly through the states $\mathbf{X}$. In (b) we substitute the product of experts and in (c) we analytically integrate out the state derivatives. We rewrite the ODE $\mathbf{f}_k$ as a linear combination of the ODE parameters (i.e. $\mathbf{B}_{\boldsymbol{\theta} k} \boldsymbol{\theta} + \mathbf{b}_{\boldsymbol{\theta} k} \overset{!}{=} \mathbf{f}_k(\mathbf{X}, \boldsymbol{\theta})$) in (d) and in (e) we normalize each factor w.r.t. $\boldsymbol{\theta}$. In (f) we normalize the product of Gaussians where mean and covariance are given by:

$$
\mathbf{r}_{\boldsymbol{\theta}} := \boldsymbol{\Omega}_{\boldsymbol{\theta}} \sum_k \mathbf{B}_{\boldsymbol{\theta} k}^T \boldsymbol{\Lambda}_k (\mathbf{m}_k - \mathbf{b}_{\boldsymbol{\theta} k}), \qquad \boldsymbol{\Omega}_{\boldsymbol{\theta}}^{-1} := \sum_k \mathbf{B}_{\boldsymbol{\theta} k}^T \boldsymbol{\Lambda}_k \mathbf{B}_{\boldsymbol{\theta} k}.
$$

The optimal variational parameters for the proxy distribution of ODE parameters are therefore given by:

$$
\hat{\boldsymbol{\lambda}} := \begin{pmatrix} \mathbb{E}_Q \boldsymbol{\Omega}_{\boldsymbol{\theta}}^{-1} \mathbf{r}_{\boldsymbol{\theta}} \\ -\frac{1}{2} \mathbb{E}_Q \boldsymbol{\Omega}_{\boldsymbol{\theta}}^{-1} \end{pmatrix} = \begin{pmatrix} \mathbb{E}_Q \sum_k \mathbf{B}_{\boldsymbol{\theta} k}^T \boldsymbol{\Lambda}_k (\mathbf{m}_k - \mathbf{b}_{\boldsymbol{\theta} k}) \\ -\frac{1}{2} \mathbb{E}_Q \sum_k \mathbf{B}_{\boldsymbol{\theta} k}^T \boldsymbol{\Lambda}_k \mathbf{B}_{\boldsymbol{\theta} k} \end{pmatrix}
$$

Similarly, for the latent states the true conditional distribution is given by:

$$
\begin{aligned}
p(\mathbf{x}_u \mid \boldsymbol{\theta}, \mathbf{X}_{/\{\mathbf{x}_u\}}, \mathbf{Y}, \boldsymbol{\phi}, \boldsymbol{\gamma}) &= Z_u^{-1}(\boldsymbol{\theta}) \prod_k \mathcal{N}\left(\mathbf{f}_k(\mathbf{X}, \boldsymbol{\theta}) \mid \mathbf{m}_k, \boldsymbol{\Lambda}_k^{-1}\right) \mathcal{N}\left(\mathbf{x}_u \mid \boldsymbol{\mu}_u(\mathbf{Y}), \boldsymbol{\Sigma}_u\right) \\
&\overset{(g)}{=} Z_u^{-1}(\boldsymbol{\theta}) \prod_k \mathcal{N}\left(\mathbf{B}_{uk} \mathbf{x}_u + \mathbf{b}_{uk} \mid \mathbf{m}_k, \boldsymbol{\Lambda}_k^{-1}\right) \mathcal{N}\left(\mathbf{x}_u \mid \boldsymbol{\mu}_u(\mathbf{Y}), \boldsymbol{\Sigma}_u\right) \\
&= Z_u^{\prime -1}(\boldsymbol{\theta}) \prod_k \mathcal{N}\left(\mathbf{x}_u \mid \left(\mathbf{B}_{uk}^T \boldsymbol{\Lambda}_k \mathbf{B}_{uk}\right)^{-1} \mathbf{B}_{uk}^T \boldsymbol{\Lambda}_k (\mathbf{m}_k - \mathbf{b}_{uk}), \left(\mathbf{B}_{uk}^T \boldsymbol{\Lambda}_k \mathbf{B}_{uk}\right)^{-1}\right) \\
&\quad \mathcal{N}\left(\mathbf{x}_u \mid \boldsymbol{\mu}_u(\mathbf{Y}), \boldsymbol{\Sigma}_u\right) \\
&= \mathcal{N}\left(\mathbf{x}_u \mid \mathbf{r}_u, \boldsymbol{\Omega}_u\right),
\end{aligned}
$$

where $Z_u(\boldsymbol{\theta})$ and $Z_u'(\boldsymbol{\theta})$ normalize the distributions. For a partially observed system, the mean $\boldsymbol{\mu}_u(\mathbf{Y})$ and covariance $\boldsymbol{\Sigma}_u$ are given by $\boldsymbol{\mu}_u(\mathbf{Y}) := \sigma^{-2} \left(\sigma^{-2} \mathbf{A}^T \mathbf{A} + \mathbf{C}_{\boldsymbol{\phi}}\right)^{-1} \mathbf{A}^T \mathbf{Y}$ and $\boldsymbol{\Sigma}_u^{-1} :=$

$\sigma^{-2}\mathbf{A}^T\mathbf{A} + \mathbf{C}_\phi^{-1}$, with matrix $\mathbf{A}$ accommodating for unobserved states by encoding the linear relationship between latent states and observations (i.e. $\mathbf{Y} = \mathbf{AX} + \mathbf{E}$, $\mathbf{E} \sim \mathcal{N}(\mathbf{0}, \sigma^2\mathbf{I})$). Once more, in (g) we define $\mathbf{B}_{uk}$ and $\mathbf{b}_{uk}$ such that the ODE $\mathbf{f}_k$ is expressed as a linear combination of the state $\mathbf{x}_k$ (i.e. $\mathbf{B}_{uk}\mathbf{x}_u + \mathbf{b}_{uk} \overset{!}{=} \mathbf{f}_k(\mathbf{X}, \boldsymbol{\theta})$). The mean and covariance are given by:

$$\mathbf{r}_u := \boldsymbol{\Omega}\left(\sum_k \mathbf{B}_{uk}^T\boldsymbol{\Lambda}_k(\mathbf{m}_k - \mathbf{b}_{uk}) + \boldsymbol{\Sigma}_u^{-1}\boldsymbol{\mu}_u(\mathbf{Y})\right), \qquad \boldsymbol{\Omega}_u^{-1} := \sum_k \mathbf{B}_{uk}^T\boldsymbol{\Lambda}_k\mathbf{B}_{uk} + \boldsymbol{\Sigma}_u^{-1}.$$

The optimal variational parameters for the proxy distribution of ODE parameters are therefore given by:

$$\hat{\boldsymbol{\psi}}_u := \begin{pmatrix} \mathbb{E}_Q\boldsymbol{\Omega}_u^{-1}\mathbf{r}_u \\ -\frac{1}{2}\mathbb{E}_Q\boldsymbol{\Omega}_u^{-1} \end{pmatrix} = \begin{pmatrix} \mathbb{E}_Q\sum_k \mathbf{B}_{uk}^T\boldsymbol{\Lambda}_k(\mathbf{m}_k - \mathbf{b}_{uk}) + \boldsymbol{\Sigma}_u^{-1}\boldsymbol{\mu}_u(\mathbf{Y}) \\ -\frac{1}{2}\mathbb{E}_Q\sum_k \mathbf{B}_{uk}^T\boldsymbol{\Lambda}_k\mathbf{B}_{uk} + \boldsymbol{\Sigma}_u^{-1} \end{pmatrix}.$$

# 8 Additional Experiments

## 8.1 Protein Signalling Transduction Pathway

Throughout this section we use different colors to denote the state dynamics and ODE parameter estimators of the methods mean-field gradient matching (blue), AGM (green), Bayesian numerical integration (purple) and Bayesian numerical integration initialized by estimates obtain from mean-field gradient matching (yellow).

### 8.1.1 Gaussian Corrupted Data with Variance 0.01

Figure 6 shows the estimation of state dynamics as well as ODE parameters for simulated data of the protein transduction model with additive Gaussian noise with variance 0.01. All methods, including mean-field gradient matching, AGM, Bayesian numerical estimation estimate the state dynamics well. Mean-field gradient matching demonstrates robustness against model misspecification since it estimates all state dynamics and ODE parameters correctly except for $k_2$ and at the same time requires a considerably lower runtime than AGM.

Figure 6: Comparison of estimation techniques by mean-field gradient matching (blue), AGM (green) and Bayesian numerical integration (purple) for the state dynamics and ODE parameters in the protein transduction model. Simulated data was corrupted with additive Gaussian noise with a variance of 0.01. Box-plots show the variance of the estimators over ten repeated experiments under the same conditions.

### 8.1.2 Gaussian Corrupted Data with Variance 0.1

The estimation of state dynamics as well as ODE parameter inference is shown in figure 7 for simulated data of the protein transduction model with additive Gaussian noise with variance 0.1. All methods, including mean-field gradient matching, AGM, Bayesian numerical estimation estimate the state dynamics well. In contrast to AGM, mean-field gradient matching and Bayesian numerical integration yield more robust parameter estimates since those estimates are similar to the ones obtained as in a similar experiment with a lower noise level (figure 6). Once more, the runtime of mean-field gradient matching is considerably lower than AGM.

Figure 7: Comparison of estimation techniques by mean-field gradient matching (blue), AGM (green) and Bayesian numerical integration (purple) for the state dynamics and ODE parameters in the protein transduction model. Simulated data was corrupted with additive Gaussian noise with a variance of 0.1. Box-plots show the variance of the estimators over ten repeated experiments under the same conditions.

### 8.1.3 Unobserved States S and Sd, Gaussian Corrupted Data with Variance 0.01

Figure 8 compares different estimators for the state dynamics and ODE parameters in the protein transduction model with unobserved states $S$ and $S_d$. Despite the deliberate model misspecification mean-field variational inference yields relatively good estimates of both state dynamics compared to AGM and even Bayesian numerical integration and requires a significantly lower runtime than AGM.

Figure 8: Comparison of estimation techniques by mean-field gradient matching (blue), AGM (green) and Bayesian numerical integration (purple) for the state dynamics and ODE parameters in the protein transduction model with unobserved states $S$ and $S_d$. Simulated data was corrupted with additive Gaussian noise with a variance of 0.01. Box-plots show the variance of the estimators over ten repeated experiments under the same conditions. "Bayes num. int mf" (yellow) denotes Bayesian numerical integration initialized with state dynamics and parameter estimates obtained from mean-field gradient matching.

### 8.1.4 Unobserved States S and Sd, Gaussian Corrupted Data with Variance 0.1

Figure 9 compares different estimators for the state dynamics and ODE parameters in the protein transduction model with unobserved states $S$ and $S_d$. Despite the deliberate model misspecification mean-field variational inference (blue) yields relatively good estimates of both state dynamics compared to AGM and even Bayesian numerical integration and requires a significantly lower runtime than AGM.

Figure 9: Comparison of estimation techniques by mean-field gradient matching (blue), AGM (green) and Bayesian numerical integration (purple) for the state dynamics and ODE parameters in the protein transduction model with unobserved states $S$ and $S_d$. Simulated data was corrupted with additive Gaussian noise with a variance of 0.1. Box-plots show the variance of the estimators over ten repeated experiments under the same conditions. "Bayes num. int mf" (yellow) denotes Bayesian numerical integration initialized with state dynamics and parameter estimates obtained from mean-field gradient matching.

## 8.2 Lorenz 96 System

The Lorenz 96 system is a minimalistic weather model that can be scaled arbitrarily as mentioned previously in section 5.3 where the particular form of ODE's are shown (equation (18)). Figure 10 demonstrates mean-field gradient matching for simultaneous parameter and state estimation in the Lorenz 96 system with a total of 1000 states and one third of randomly chosen states remaining unobserved. Our mean-field gradient matching method demonstrates good simultaneous parameter and state estimation.

Figure 10: Mean-field gradient matching is used for parameter and state estimation for the Lorenz 96 system with a total of 1000 states. The first 8 of 1000 states are shown where one third of randomly chosen states are unobserved. Red bars and curves respectively denote the true parameters and trajectories and purple denotes their estimation. Both the ODE parameter $\theta$ (top left-hand side) as well as the states are estimated well.

## 8.3 FitzHugh-Nagumo System

Parameter inference for the FitzHugh-Nagumo system using gradient matching was already studied by Macdonald et al. [2015]. In this section we investigate the parameter inference using our mean-field gradient matching approach with the same experimental setup as in Macdonald et al. [2015]. The ODE's for the FitzHugh-Nagumo system are given by:

$$\dot{V} = \psi \left( V - \frac{V^3}{3} - R \right), \qquad \dot{R} = -\frac{1}{\psi} \left( V - \alpha + \beta R \right),$$

where $\alpha$ and $\beta$ are the ODE parameters and we assume that the parameter $\psi$ is given and set to $\psi = 3$. Notice that applying our mean-field gradient matching is not entirely straightforward since, although the ODE's are linear in the parameters $\alpha$ and $\beta$ and the state $R$, the ODE's are *not* linear in the state $V$. To circumvent this we fix the state $V$ to a GP fit through observations of the state $V$. In other words, after fixing the state $V$ to it's GP fit, we don't re-estimate the state in the mean-field coordinate ascent framework.

Figure 11 demonstrates our mean-field gradient matching method for the fully observable system as shown in figure 11. The computational time for the estimation is one second.

Figure 11: The FitzHugh-Nagumo system is fully observed with an SNR=10. Red bars and curves respectively denote the true parameters and trajectories and purple denotes their estimation. Black curves show the trajectories obtained by numerical integration with the estimated ODE parameters. The ODE parameters $\alpha$ and $\beta$ are estimated well (left-hand side plot) using mean-field gradient matching. Error bars indicate one standard deviation.

## 8.4 Glucose Uptake in Yeast

The ODE's governing the glucose uptake in yeast are described by mass-action kinetics. We use the same notation as Schillings et al. [2015] to describe the ODE's:

$$\dot{x}^e_{Glc} = -k_1 x^e_E x^e_{Glc} + k_{-1} x^e_{E-Glc}$$
$$\dot{x}^i_{Glc} = -k_2 x^i_E x^i_{Glc} + k_{-2} x^i_{E-Glc}$$
$$\dot{x}^i_{E-G6P} = k_4 x^i_E x^i_{G6P} - k_{-4} x^i_{E-G6P}$$
$$\dot{x}^i_{E-Glc-G6P} = k_3 x^i_{E-Glc} x^i_{G6P} - k_{-3} x^i_{E-Glc-G6P}$$
$$\dot{x}^i_{G6P} = -k_3 x^i_{E-Glc} x^i_{G6P} + k_{-3} x^i_{E-Glc-G6P} - k_4 x^i_E x^i_{G6P} + k_{-4} x^i_{E-Glc}$$
$$\dot{x}^i_{E-Glc} = \alpha \left( x^i_{E-Glc} - x^e_{E-Glc} \right) + k_1 x^e_E x^e_{Glc} - k_{-1} x^e_{E-Glc}$$
$$\dot{x}^i_{E-Glc} = \alpha \left( x^e_{E-Glc} - x^i_{E-Glc} \right) - k_3 x^i_{E-Glc} x^i_{G6P} + k_{-3} x^i_{E-Glc-G6P} + k_2 x^i_E x^i_{Glc} - k_{-2} x^i_{E-Glc}$$
$$\dot{x}^e_E = \beta \left( x^i_E - x^e_E \right) - k_1 x^e_E x^e_{Glc} + k_{-1} x^e_{E-Glc}$$
$$\dot{x}^i_E = \beta \left( x^e_E - x_E \right) - k_4 x^i_E x^i_{G6P} + k_{-4} x^i_{E-G6P} - k_2 x^i_E x^i_{Glc} + k_{-2} x^i_{E-Glc}$$

where $k_1$, $k_{-1}$, $k_2$, $k_{-2}$, $k_3$, $k_{-3}$, $k_4$, $k_{-4}$, $\alpha$ and $\beta$ are the ODE parameters. Notice that, due to the mass-action kinetics form of the ODE's, the ODE's are linear in the parameters as well as linear in a *single* state. We can therefore readily apply our mean-field gradient matching method to estimate the parameters which is shown in figure 12 for the fully observed system and in figure 13 for the indirectly observed system. The computational time for both estimations is ten seconds.

Figure 12: Mean-field gradient matching is used to estimate the ODE parameters of the fully observed glucose uptake into yeast. Red bars and curves respectively denote the true parameters and trajectories and purple denotes their estimation. Black curves show the trajectories obtained by numerical integration with the estimated ODE parameters. Although the ODE parameters are not estimated perfectly, the state trajectories obtained by numerical integration with the estimated parameters (black curves) approximate the true trajectories well. The trajectory of the state $x_E^i$ is not shown.

Figure 13: Mean-field gradient matching is used to estimate the ODE parameters of the indirectly observed glucose uptake into yeast where the states $x_{Glc}^e$, $x_{Glc}^i$ and $x_{E-G6P}^i$ are indirectly observed through the combination $(x_{Glc}^e + x_{Glc}^i + x_{E-G6P}^i)/3$. Red bars and curves respectively denote the true parameters and trajectories and purple denotes their estimation. Black curves show the trajectories obtained by numerical integration with the estimated ODE parameters. The trajectory of the state $x_E^i$ is not shown.

### 8.4.1 Dynamic Causal Models

Dynamic causal models (DCM) propose that the activity between neuronal populations are governed by ODE's whose parameters and particular form can give insight into the mechanism behind neurodegenerative diseases. The nonlinear ODE's for DCM are given by:

$$\dot{x} = \left( \mathbf{A} + \sum_{i=1}^{m} u_i \mathbf{B}^{(i)} + \sum_{j=1}^{m} x_j \mathbf{D}^{(j)} \right) x + \mathbf{C}u \qquad \text{neuronal activity}$$

$$\dot{s} = x - \kappa s - \gamma(f - 1) \qquad \text{vasosignalling}$$

$$\dot{f} = s \qquad \text{blood flow induction}$$

$$\dot{v} = \tau^{-1} \left( f - v^{1/\alpha} \right) \qquad \text{blood volume changes}$$

$$\dot{q} = \tau^{-1} \left( fE(f, E_0)/E_0 - v^{1/\alpha}q/v \right) \qquad \text{deoxyhemoglobin changes}$$

where the matrices $\mathbf{A}$, $\mathbf{B}$, $\mathbf{D}$ and $\mathbf{C}$ control the endogenous neuronal couplings, the modulation of connectivity by external inputs $u$, the nonlinear modulation and the driving inputs, respectively. The remaining parameters $\kappa$, $\gamma$, $\alpha$, $\tau$ and $E_0$ are hemodynamic parameters which we treat as given. Notice that the DCM ODE's are linear in the neuronal parameters $\mathbf{A}$, $\mathbf{B}$, $\mathbf{C}$ and $\mathbf{D}$ and linear in a *single* neuronal state $x$ as well as in the vasosignalling $s$. In order to apply our mean-field gradient matching method we treat only the neuronal states $x$ and the vasosignalling $s$ as unobserved and the remaining states $f$, $v$ and $q$ as observed. Similar to the FitzHugh-Nagumo system, we can therefore fix the states $f$, $v$ and $q$ by a GP fit through the observations of the states $f$, $v$ and $q$. The estimation of neuronal parameters (i.e. neuronal couplings) as well as the neuronal state trajectories are shown in figure 14 for the visual attention model (i.e. three state system) with the true parameters taken from figure 7 in Stephan et al. [2008]. The computational time for the estimation is ten seconds.

Figure 14: Mean-field gradient matching is used to estimate the ODE parameters of DCM for the visual attention system. The true ODE parameters were taken from figure 7 in Stephan et al. [2008]. Red bars and curves denote respectively the true parameters and trajectories and purple denotes their estimation. Black curves show the trajectories obtained by numerical integration with the estimated ODE parameters. The parameters $a_{(\cdot,\cdot)}$, $b_{(\cdot,\cdot,\cdot)}$, $c_{(\cdot,\cdot)}$ and $d_{(\cdot,\cdot,\cdot)}$ denote the entries of the matrices $\mathbf{A}$, $\mathbf{B}^{(\cdot)}$, $\mathbf{C}$ and $\mathbf{D}^{(\cdot)}$, respectively. The first subscript gives the row index, the second the column index and the third the matrix sheet for matrices $\mathbf{B}$ and $\mathbf{D}$. The remaining entries of each matrix are set to zero and are not estimated. Although the ODE parameters are not estimated perfectly, the neuronal state trajectories obtained by numerical integration with the estimated parameters (black curves) approximate the true trajectories well. No priors were placed directly on the parameters except for the self-inhibitory parameters $a_{11}$, $a_{22}$ and $a_{33}$.

Contrary to existing methods for parameter estimation in DCM, we obtain reasonable parameter estimates despite *not* placing any direct priors on the ODE parameters except of the self-inhibitory parameters $a_{11}$, $a_{22}$ and $a_{33}$.

### 8.4.2 Kernels

Gaussian process priors assigned to the various systems investigated in this paper, namely Lotka-Volterra, protein signalling transduction pathway, Lorenz 96, FitzHugh Nagumo, dynamic causal models and glucose uptake into yeast, are shown in figure 15.

Figure 15: Samples from the Gaussian process priors assigned to the various dynamical systems are shown. Colored lines simply highlight randomly chosen samples. The radial basis function (rbf) kernel was assigned to four systems and the sigmoid kernel to two other systems.

Ideally, the Gaussian process prior should approximate the functional form of the states. The radial basis function (rbf) kernel is therefore suitable for the Lotka-Volterra-, Lorenz 96-, FitzHugh-Nagumo-and Dynamic Causal Models systems. The sigmoid kernel is more suited for the protein signalling transduction pathway and the glucose uptake in yeast system that exhibit a more steady-state behaviour over time.