[Reviews · NeurIPS 2017]

Reviewer 1



he paper concerns identification of nonlinear ODEs. The method combines some previous ideas (Calderhead, Dondelinger) with a variational approximation. A mean-field approximation over the states and parameters is assumed. Standard mean-field updates are derived. On a Lotka-Volterra model, the method outperforms the state-of-the-art, and on a problem with many hidden states a solution can be found (presumably this is beyond the capabilities of the state of the art). I confess that I found this paper a challenging read. There is quite a bit of background material, which the authors do seem to cover in some depth, but nonetheless the space constraints make some explanations rather curt. I suggest annotating the equation between 7 and 8 to explain which terms come from where. The mixture-of-experts approach to combining the GP prior and the ODE seems to be a crucual part of the background material. If I understand correctly, a previous approach (Dondelinger) sampled the resulting distribution. This paper proposes a variational mean-field approximation. Since in general one might expect a variational approximation to perform *worse* than MCMC, the results in section 5.1 are rather surprising: the variational approximation appears to fit better than MCMC-based approaches. Without analysis and comment on this, the experiment feels rather unfinished. Another confusing aspect of the paper is the treatment of the parameters \theta. eq. 10 implies that the authors are intending to estimate the parameters via MAP, integrating over the states X. However the equations between 12 and 13 imply that a mean-field posterior is assumed for theta. I applaud the authors for stating up-front the class of problems for which their approximation applies. It seems like good-science top be upfront about this. However, it's not clear to me why the method should be restricted to this case, or how it might be extended to further cases. I feel that this aspect of the paper needs more explanation. In its current form, I can;t recommend this paper for acceptance at NIPS. I think there's some valuable work in here, but the authors need to expplain their ideas, the background and the results with more clarity. *** POST DISCUSSION EDIT *** I have read the author rebuttal and increased my score. The authors have identified parts of their paper to clarify, and having seen also the other reviews I think the paper makes a good contribution. Despite some clarity flaws, the paper continues an important line of work and deserves discussion in the NIPS community.

Reviewer 2



This paper proposes variational inference for models involving dynamical systems whose dynamics is described by a set of differential equations. The paper builds upon previous work on gradient matching and Gaussian processes (GPs); the idea is to interpolate between observations using GPs to obtain approximate solutions that when plugged back into the differential equations lead to some nice properties (derivatives of GPs are GPs) that can be exploited to carry out optimization of differential equation parameters without the need to solve the system explicitly. This work extends previous work in this domain by proposing variational inference for the states and optimization of the lower bound to estimate differential equation parameters. While the paper reports some interesting improvements with respect to previous gradient matching approaches and the paper is interesting and well written, I think that the lack of novelty is a major issue. Gradient matching has been extensively studied in various works referenced in the paper, so there is not much novelty from the modeling perspective. Also, the proposed inference framework is a rather straightforward application of mean field variational inference, so again there is no particular innovation apart from the specialization of this particular inference technique to this model. ** After rebuttal In light of the rebuttal and the comments from the other reviewers, I've decided to increase the score to 6.

Reviewer 3



This paper proposes a mean-field method for joint state and parameter inference of ODE systems. This is a problem that continues to receive attention because of its importance in many fields (since ODEs are used so widely) and the limitations of existing gradient-based methods. The solution described in the paper is said to outperform other approaches, be more applicable, and scale better. The paper lays out its contribution clearly in the introduction. It does a good job of discussing the merits and drawbacks of previous approaches, and explaining how it fits in this context. The proposed methodology appears sound (although it might be useful to explain some of the derivations in a little more detail) and the results indicate that it performs much better than the state of the art. The main advantage of the new method is its execution time, which is shown to be orders of magnitude faster than methods with comparable (or worse) accuracy. In particular, the scalability analysis appears very promising for the applicability of this method to larger models. Overall, I found this to be a strong paper. My comments mainly concern the presentation, and most are minor. - The biggest comment has to do with the derivations of the evidence lower bounds and Equation (16). I believe it would be useful to have some more details there, such as explicitly pointing out which properties they rely on. It is also not fully clear to me how the E-M steps of Algorithm 1 make use of the previous derivations (although that may be because I have limited experience in this field). There appears to be some space available for expansion but, if necessary, I suppose that sections 2 and 3 could be combined. - Besides that, there is a statement that appears contradictory. Lines 82-86 describe how parameters and state are alternately sampled in Calderhead et al. and Dondelinger et al. However, lines 29-30 make it seem like these approaches cannot infer both state and parameters. I would suggest either to rephrase (perhaps remove the "simultaneously"?) or explain why this is a problem. - As far as the experiments are concerned, I think it would be interesting to run the Lotka-Volterra example of Section 5.1 for longer, to confirm that the inferred state displays periodicity (perhaps comparing with the other methods as well). However, I believe the results do a good job of supporting the authors' claims. Minor comments: -------------- l2: grid free -> grid-free (also later in the text) l52: the phrasing makes it sound like the error has zero mean and zero variance; perhaps rephrase to clarify that the variance can be generally elliptical / different for each state? Equations 3 - 8: There is a small change in the notation for normal distributions between Eqs 3-4 and 5-8. For the sake of clarity, I think it is worth making it consistent, e.g. by changing the rhs of (3) to N(x_k | 0, C) instead of N(0,C) (or the other way around). l71: What is the difference between C' and 'C ? Equation 9: no upper bound on the sum l99: contains -> contain l99: "arbitrary large" makes me think that a sum term can include repetitions of an x_j (to effectively have power terms, as you would have in mass-action kinetics with stoichiometry coefficients > 1), but it is not fully clear. Is this allowed? (in other words, is M_{ki} a subset of {1,...,K}, or a multiset of its elements?) l139: what does "number of states" refer to here? the number of terms in the sum in Eq 9? Figure 1: in the centre-right plot, it seems like the estimate of \theta_4 is missing for the spline method l155: Predator is x_2, prey is x_1 l163: move the [ after the e.g. : "used in e.g. [Calderhead et al. ..." l168: usining -> using l171: I would add commas around "at roughly 100 times the runtime" to make the sentence read easier l175: partial -> partially l187: (3) -> (Figure 3) l195-196: Figure 2 does not show the noise changing; I think having two references would be more helpful: "[...] if state S is unobserved (Figure 2) and [...] than our approach (Figure 3)." l198: perhaps "error" is better than "bias" here? Unless it is known how exactly the misspecification biases the estimation? l202: Lorentz -> Lorenz (and later) l207: stochstic -> stochastic l208: number states -> number of states l209: Due the dimensionality -> Due to the dimensionality, l212: equally space -> equally spaced l214: correct inferred -> correctly inferred Figure 4: In the last two lines of the caption: wheras -> whereas; blue -> black (at least that's how it looks to me); is corresponding to -> corresponds to l219: goldstandard -> gold standard l253: modemodel -> modelling